# Twitter and Facebook posts about COVID-19 are less likely to spread misinformation compared to other health topics

**David A. Broniatowski**[1,2]*, **Daniel Kerchner**[3], **Fouzia Farooq**[4], **Xiaolei Huang**[5], **Amelia M. Jamison**[6¤], **Mark Dredze**[7], **Sandra Crouse Quinn**[6], **John W. Ayers**[8]

**1** Department of Engineering Management and Systems Engineering, School of Engineering and Applied Science, The George Washington University, Washington, DC, United States of America, **2** Institute for Data, Democracy and Politics, The George Washington University, Washington, DC, United States of America, **3** George Washington University Libraries, The George Washington University, Washington, DC, United States of America, **4** Department of Epidemiology, Graduate School of Public Health, University of Pittsburgh, Pittsburgh, PA, United States of America, **5** Department of Computer Science, University of Memphis, Memphis, TN, United States of America, **6** Department of Family Science, Center for Health Equity, School of Public Health, University of Maryland, College Park, MD, United States of America, **7** Department of Computer Science, Whiting School of Engineering, Johns Hopkins University, Baltimore, MD, United States of America, **8** Division of Infectious Diseases and Global Public Health, University of California San Diego, La Jolla, CA, United States of America

¤ Current address: Department of Health, Behavior and Society, Bloomberg School of Public Health, Johns Hopkins University, Baltimore, MD, United States of America
* broniatowski@gwu.edu

**Data Availability Statement:** Data are available at Harvard's Dataverse at this link: https://doi.org/10.7910/DVN/X6AF8I; however, there are some legal restrictions on sharing data, per the terms of service of social media platforms and NewsGuard's

## Abstract

The COVID-19 pandemic brought widespread attention to an "infodemic" of potential health misinformation. This claim has not been assessed based on evidence. We evaluated if health misinformation became more common during the pandemic. We gathered about 325 million posts sharing URLs from Twitter and Facebook during the beginning of the pandemic (March 8-May 1, 2020) compared to the same period in 2019. We relied on source credibility as an accepted proxy for misinformation across this database. Human annotators also coded a subsample of 3000 posts with URLs for misinformation. Posts about COVID-19 were 0.37 times as likely to link to "not credible" sources and 1.13 times *more* likely to link to "more credible" sources than prior to the pandemic. Posts linking to "not credible" sources were 3.67 times more likely to include misinformation compared to posts from "more credible" sources. Thus, during the earliest stages of the pandemic, when claims of an infodemic emerged, social media contained proportionally less misinformation than expected based on the prior year. Our results suggest that widespread health misinformation is not unique to COVID-19. Rather, it is a systemic feature of online health communication that can adversely impact public health behaviors and must therefore be addressed.

licensing terms as follows: · Twitter o Link to Terms of Service: https://developer.twitter.com/en/developer-terms/agreement-and-policy o Availability: Tweets IDs are now available on Harvard's Dataverse at https://doi.org/10.7910/DVN/X6AF8I · CrowdTangle o Relevant Terms of Service: CrowdTangle prohibits providing raw data to anyone outside of a CrowdTangle user's account. The user can share the findings, but not the data. If a journal asks for data to verify findings, the CrowdTangle user may send a .csv, but it cannot be posted publicly, and the journal must delete it after verification. o Availability: Frequency counts of each top-level domain in this dataset are now available on Harvard's dataverse at https://doi.org/10.7910/DVN/X6AF8I. Additionally, CrowdTangle list IDs are provided in the references. Anyone with a CrowdTangle account may access these lists and the corresponding raw data. Researchers may request CrowdTangle access at https://help.crowdtangle.com/en/articles/4302208-crowdtangle-for-academics-and-researchers · Domain frequency counts for Twitter and Facebook o Relevant Terms of Service: None o Availability: csv files containing the frequencies of each domain in each dataset are now available at https://doi.org/10.7910/DVN/X6AF8I · MediaBiasFactCheck o Link to Terms of Service: https://mediabiasfactcheck.com/terms-and-conditions/ o Availability: MediaBiasFactCheck ratings are currently publicly available at https://mediabiasfactcheck.com/ · NewsGuard o Relevant Terms of Service: https://www.newsguardtech.com/terms-of-service/ o Availability: These data were provided under license for a fee by a third party provider (NewsGuard). Researchers seeking to use NewsGuard data in their own research may inquire about licensing the data directly from NewsGuard for a fee. Researcher licensing information can be found here: https://www.newsguardtech.com/newsguard-for-researchers/ All DOIs provided above are activated and publicly accessible. The authors did not receive any special privileges in accessing any of the third-party data that other researchers would not have.

**Funding:** This work was supported in part by grant number R01GM114771 to D.A. Broniatowski and S.C. Quinn, and by the John S. and James L. Knight Foundation to the GW Institute for Data, Democracy, and Politics. The funders had no role in study design, data collection and analysis, decision to publish, or preparation of the manuscript.

**Competing interests:** David A. Broniatowski received an honorarium from the United Nations Shot@Life Foundation – a non-governmental

## Introduction

On February 15, 2020, the Director General of the World Health Organization declared that the coronavirus disease 2019 pandemic (COVID-19) spurred an "infodemic" of misinformation [1]. This claim quickly became accepted as a matter of fact among government agencies, allied health groups, and the public at-large [2–10]. For instance, during the past year over 15,000 news reports archived on Google News refer to a COVID-19 "infodemic" in their title and about 5,000 scholarly research reports on Google Scholar refer to an infodemic in the title and/or abstract. Despite this widespread attention, the claim that online content about COVID-19 is more likely to be false than other topics has not been tested.

We seek to characterize the COVID-19 infodemic's scale and scope in comparison to other health topics. In particular, we focus on the opening stages of the infodemic–March through May, 2020 –when case counts began to increase worldwide, vaccines were not yet available, and concerted collective action–such as social distancing, mask-wearing, and compliance with government lockdowns–was necessary to reduce the rate at which COVID-19 spread. Misinformation during this time period was especially problematic because of its potential to undermine these collective efforts. Our study therefore aims to answer the following question:

- Were posts about COVID-19 more likely to contain links to misinformation when compared to other health topics?

Beyond the sheer volume of links shared, one might define an "infodemic" by the likelihood that a particular type of post might go viral. Thus, our second question:

- When it comes to COVID-19, were links containing misinformation more likely to go viral?

To answer these questions, we must rely on a scalable method. One commonly used proxy for misinformation is *source credibility*. If the infodemic was indeed characterized by false content, one might expect a higher proportion of this content to come from low credibility sources that "lack the news media's editorial norms and processes for ensuring the accuracy and credibility of information" [11]. Thus, our third question:

- Does content from less credible sources include more misinformation?

## Evidence before this study

Prior studies [12] found that low-credibility content was, in fact, rare on Twitter, albeit shared widely within concentrated networks [13]. We only found two studies comparing across multiple social media platforms [13, 14], with both studies concluding that the prevalence of low-credibility content varied significantly between platforms. None of these studies compared COVID-19 content to other health topics.

To our knowledge, this study is the first to evaluate the claim of an infodemic by comparing COVID-19 content to other health topics. We analyzed hundreds of millions of social media posts to determine if COVID-19 posts pointed to lower-credibility sources compared to other health content.

## Materials and methods

### Data collection

Data comprised all public posts made to Twitter, and public posts from Facebook from pages (intended to represent brands and celebrities) with more than 100,000 likes, and groups

organization that promotes childhood vaccination. Mark Dredze holds equity in Sickweather Inc. and has received consulting fees from Bloomberg LP and Good Analytics Inc. None of these organizations had any role in the study design, data collection, and analysis, decision to publish, or preparation of the article. The remaining authors declare no competing interests.

(intended as venues for public conversation) with at least 95,000 members or those based in the US with at least 2,000 members.

**COVID-19 tweets.** First, we collected English language tweets from Twitter matching keywords pertaining to COVID-19 [15] between March 8, 2020 and May 1, 2020. Next, we compared these to tweets containing keywords pertaining to other health topics [16] for the same dates in 2019.

We obtained COVID-19 tweets using the Social Feed Manager software [17], which collected English-language tweets from the Twitter API's statuses/filter streaming endpoint (https://developer.twitter.com/en/docs/tweets/filter-realtime/api-reference/post-statuses-filter) matching keywords of "#Coronavirus", "#CoronaOutbreak", and "#COVID19" posted between March 8, 2020 and May 1, 2020 [15].

**Health tweets.** We obtained tweets about other health topics using the Twitter Streaming API to collect English-language tweets containing keywords pertaining to generalized health topics posted between March 8, 2019 and May 1, 2019 (keywords are listed in reference [16]).

**Facebook data.** Next, we collected comparable data from Facebook for the same dates using CrowdTangle [18]–a public insights tool owned and operated by Facebook. Specifically, we collected English-language posts from Facebook Pages matching keywords of:

- "coronavirus", "coronaoutbreak", and "covid19" posted between March 8, 2020 and May 1, 2020, downloaded on June 2–3, 2020.

- the same health-related keywords used in the health stream posted between March 8, 2019 and May 1, 2019, downloaded on July 13–14, 2020.

**Ethics.** The data used in this article are from publicly available online sources, the uses of which are deemed exempt by the George Washington University institutional review board (180804).

## Credibility categorization

Our analysis draws upon an assumption that is widespread in prior work [11–14, 19, 20]: that "the attribution of 'fakeness' is . . . not at the level of the story but at that of the publisher." [19]. This assumption is attractive because it is scalable, allowing researchers to analyze vast quantities of posts by characterizing their source URLs. We therefore extracted all Uniform Resource Locators (URLs) in each post. We used the "tldextract" Python module [21] to identify each URL's top-level domain (for example the top-level domain for http://www.example.com/this-is-an-example-article.html is example.com), unshortening links (e.g., "bit.ly/x11234b") if necessary (see Appendix A in S1 File). We grouped these top-level domains into three categories reflecting their overall credibility using a combination of credibility scores from independent sources (NewsGuard; http://www.newsguard.com/, and MediaBiasFactCheck; http://www.MediaBiasFactCheck.com), as follows (see Appendix B in S2 File for details):

**More credible.** This category contained the most credible sources. Top-level domains were considered "more credible" if they fit into one of the following two categories:

- **Government and Academic Sources,** defined by Singh et al. [12], as "high quality health sources", included official government sources such as a public health agency (e.g., if the top-level domain contained .gov), or academic journals and institutions of higher education (e.g., if the top-level domain contained .edu; see Appendix B in S2 File).

- **Other More Credible Sources,** defined by Singh et al. [12], as "traditional media sources", were given a credibility rating of at least 67% by NewsGuard, or rated as "very high" (coded as 100%) or "high" (80%) on the MediaBiasFactCheck factual reporting scale (NewsGuard

and MediaBiasFactCheck scores are strongly correlated, r = 0.81, so we averaged these scores when both were available).

**Less credible.** Top-level domains were considered "less credible" if they were given a credibility rating between 33% and 67% by NewsGuard, or rated as "mostly factual" (60%) or "mixed" (40%) on the MediaBiasFactCheck factual reporting scale (averaging these when both were available).

**Not credible.** These sources contained the least credible sources, such as conspiracy-oriented sites, but also government-sponsored sites that are generally considered propagandistic. Top-level domains were considered "not credible" if they:

- Were given a credibility rating of 33% or less by NewsGuard or rated as "low" (20%) or "very low" (0%) on the MediaBiasFactCheck factual reporting scale.

- Were rated as a "questionable source" by MediaBiasFactCheck.

Like prior work, [12, 13, 19, 20, 22] our analysis draws upon a widespread simplifying assumption: that "the attribution of 'fakeness' is . . . not at the level of the story but at that of the publisher." [19]. This assumption is attractive because it is scalable. However, in the interest of evaluating it for health topics, we performed an additional validity check. To determine the content of each credibility category, we developed a codebook (Table 1) to assess the presence of false claims. We generated a stratified sample of 3000 posts by randomly selecting 200 posts from each COVID-19 dataset for each credibility category (More, Less, Not Credible, Unrated) and a set of 200 "in platform" posts (i.e., those linking to Twitter from Twitter or Facebook from Facebook). Three authors (DK, FF, and AMJ) manually labeled batches of 100 posts from each platform each until annotators achieved high interrater reliability

**Table 1. Codebook for qualitative analysis.**

| | |
|---|---|
| Misinformation | **Yes** = if the message contains egregious falsehoods, conspiracy theories, or misleading use of data **related to COVID-19**. Also, news reports repeating these claims. Any story that is promoting misinformation. |
| | **No** = Anything else. |
| Uncertainty | **Yes** = if the message expresses the idea that there is a lot that science still does not know about the coronavirus and/or casts doubt on science and scientists. e.g. "how do we know that masks work?" |
| | **No** = Anything else |
| Partisan bias | **Conservative** = expresses viewpoints supporting conservative politics or opposing liberal politics **in the United States**. Includes key talking points. |
| | **Liberal** = expresses viewpoints supporting liberal politics or opposing conservative politics **in the United States**. Includes key talking points. |
| | **O**ther = expresses a political opinion but not one of the two major US parties. International politics. |
| | **None** = No political content. |
| Content Area | **Political** = primary purpose of content is political. |
| | **Lifestyle** = discusses non-medical aspects of the pandemic including societal impacts. Impacts on life—school closures, cancellations, impacts on work, travel restrictions, long lines. |
| | **Opportunistic** = using popularity of COVID to market unrelated or semi-related content. Hashtag hijacking. |
| | **Information Sharing** = link sharing or news sharing (does not need to be accurate information) |
| | **Discussion** = first person discussion of experiences or of information. |

Using this codebook, annotators achieved Krippendorff's $\alpha_1 = 0.742$ on the first set of 100 posts. Annotators achieved $\alpha_2 = 0.811$ on the second set of 100 posts. The remaining 2400 posts were then split uniformly at random between the three annotators.

(Krippendorff's α>0.80), which we obtained on the second round (α = 0.81). Disagreements were resolved by majority and ties adjudicated by a fourth author (DAB). The remaining 2400 posts were then split equally between all three annotators. We also generated qualitative descriptions for each credibility category.

*Virality analysis.* We conducted negative binomial regressions for each COVID dataset to predict the number of shares or retweets for each original post (Facebook and Twitter share counts were current as of June 2, 2020, and May 31, 2020, respectively). Following Singh et al. [12], we analyzed high-quality health sources separately from traditional media sources, separating the "more credible" category into two subcategories: "academic and government" and "other more credible" sources. For tweets with multiple URLs, we assigned each tweet with a lower-credibility URL ("not credible" or "less credible") to its least credible category (see Appendix C in S3 File).

## Results

We identified 305,129,859 posts on Twitter, 13,437,700 posts to Facebook pages, and 6,577,307 posts to Facebook groups, containing keywords pertaining to COVID-19 and other health conditions. These posts contained 41,134,540 URLs (excluding in-platform links such as retweets and shares) including 554,378 unique top-level domains. 14,609 (2.6%) of these unique top-level domains were assigned a credibility rating, these top-level domains accounted for 19,294,621 (47%) of all URLs shared. The remaining URLs were unrated (see S1 Fig for raw counts).

### Content of credibility categories

We conducted an inductive analysis of each credibility category to validate the use of credibility as a proxy for misinformation (see Table 2 for examples of URLs from each category).

**"Not credible" sources contained more misinformation than "more credible" sources.** In our stratified random sample of 3000 posts, those with URLs rated as "not credible" were 3.67 (95% CI: 3.50–3.71) times more likely to contain false claims than "more credible" sources (Fig 1). Results were comparable when comparing only those posts labeled as containing news or information (see S2 Fig), and we did not detect a significant difference between high-quality health sources (5.33% misinformation, 95% CI: 0.00–10.42, n = 75) and more credible traditional media sources (5.33% misinformation, 95% CI: 3.41–7.26, n = 525). Neither

**Table 2. Examples for each credibility category.**

| Link | Top-Level Domain | Credibility Rating | Article Headline |
|---|---|---|---|
| https://www.gov.uk/government/publications/covid-19-guidance-on-social-distancing-and-for-vulnerable-people/guidance-on-social-distancing-for-everyone-in-the-uk-and-protecting-older-people-and-vulnerable-adults | gov.uk | More Credible (Government or Academic) | Guidance on Social Distancing for Everyone in the UK |
| https://www.thehindu.com/news/national/ramp-up-testing-its-our-only-weapon-against-coronavirus-rahul-gandhi/article31354101.ece | thehindu. com | More Credible (Other) | Coronavirus \| Lockdown only a pause button, testing is the only weapon, says Rahul Gandhi |
| https://www.rappler.com/world/asia-pacific/interview-south-korean-ambassador-han-dong-man-coronavirus | rappler.com | Less Credible | Lessons from South Korea: Transparency, Rapid Testing, No Lockdowns |
| https://www.afa.net/the-stand/culture/2020/04/shutdowns-were-pointless-all-along/ | afa.net | Not Credible | Shutdowns Were Pointless All Along |

To comply with NewsGuard's terms of service, examples are drawn from websites that have been rated by MediaBiasFactCheck, but not by NewsGuard.

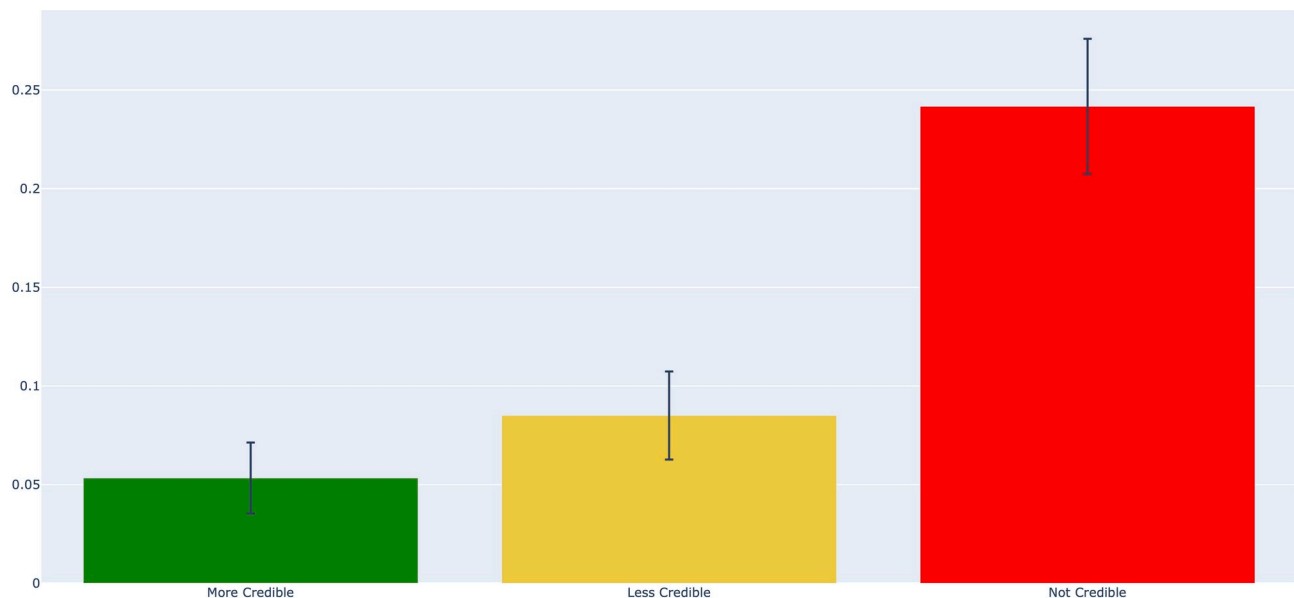

**Fig 1. Proportions of misinformation for each credibility category.** Error bars represent 95% confidence intervals.

intermediate "less credible" sources (8.50% misinformation, 95% CI: 6.11–10.89), "unrated" sources (7.33% misinformation, 95% CI: 5.10–9.56), or "in platform" sources (5.17% misinformation, 95% CI: 4.26–6.07) were statistically significantly more likely to contain misinformation when compared to "more credible" sources (5.33% misinformation, 95% CI: 3.41–7.26, n = 600).

Beyond these misinformation ratings, we calculated the proportions of each content type in our codebook, for each credibility category (S2 Fig). A qualitative description of each category follows.

**More credible.** These sources primarily shared news and government announcements. Content was rarely political, although users sometimes editorialized, often with liberal bias. Here, misinformation reported on, and potentially amplified, questionable content, such as explaining conspiracy theories or reporting on claims that bleach cures COVID. Some content also expressed uncertainty around COVID-19 science, pointing out limitations of data and models, and acknowledging major questions could not yet be answered.

**Less credible.** These sources contained a wide variety of content. Non-US politics were common, especially from Indian, Chinese, and European sources. Misinformation in this category included some political conspiracy theories, but also more subtle falsehoods including suggesting COVID is less severe than flu, promoting hydroxychloroquine as a cure, or claiming that "lockdowns" are an overreaction. This category also includes content that inadvertently amplified questionable content while attempting to debunk it.

**Not credible.** Misinformation was more common in this category. Common themes included: blaming China for the virus, questioning its origins, rejecting vaccines, and framing COVID as undermining U.S. President Trump. These sources also tended to have a conservative political bias. Content emphasizing scientific uncertainty suggested that response measures were unjustified or that science was distorted for political ends. This category also included propaganda narratives, often extolling Russian and Chinese COVID responses.

## Comparison to other health topics prior to the pandemic

Posts about COVID-19 were less likely to contain links to "not credible" sources and more likely to contain links to "more credible" sources when compared to other health topics prior to the pandemic. On average, URLs shared were more likely to be credible than non-credible during the pandemic (Fig 2). Among rated links, the proportion of "not credible" links shared during the pandemic in posts containing COVID-19 keywords was lower on Twitter (RR = 0.37; 95% CI: 0.37–0.37), Facebook Pages (RR = 0.41; 95% CI: 0.40–0.42), and Facebook Groups (RR = 0.37; 95% CI: 0.37–0.38). Additionally, the proportion of "more credible" links in posts containing COVID-19 keywords was higher on Twitter (RR = 1.13; 95% CI: 1.13– 1.13), Facebook Pages (RR = 1.07; 95% CI: 1.07–1.07), and Facebook Groups (RR = 1.03; 95% CI: 1.02–1.03). These results replicated when focusing only on "high-quality health sources"— academic and government sources—for all three platforms: Twitter (RR = 3.52; 95% CI: 3.50– 3.54), Facebook Pages (RR = 1.15; 95% CI: 1.14–1.17), and Facebook Groups (RR = 1.09; 95% CI: 1.06–1.11). URLs were also less likely to be unrated during the pandemic: Twitter RR = 0.67 (95% CI: 0.67 to 0.67), Facebook Pages RR = 0.74 (95% CI: 0.74 to 0.74), and Facebook Groups RR = 0.58 (95% CI: 0.58 to 0.58) (see Supplementary Material).

## The least credible posts are not the most viral

Even if low credibility content is less widespread on Twitter and Facebook, it can still be harmful if it garners more engagement. We therefore compared the average number of shares for each credibility category. We did not find that the least credible content was the most widely shared. Rather, on Twitter and Facebook Pages, the most viral posts contained links to government and academic sources, whereas intermediate "less credible" sources were the most viral in Facebook Groups (Fig 3).

## Discussion

Like prior studies [12, 14, 22], we find that there is indeed an overwhelming amount of content pertaining to COVID-19 online, making it difficult to discern truth from falsehood. Furthermore, we found that posts with URLs rated as "not credible" were indeed more likely to contain falsehoods than posts in other categories.

We are the first to compare this content to other health topics across platforms, adding much needed context. Upon comparison, we found that social media posts about COVID-19 were more likely to come from credible sources, and less likely to come from non-credible sources. Thus, available evidence suggests that misinformation about COVID-19 is proportionally quite rare, especially when compared to misinformation about other health topics.

Although sources rated as "not credible" were roughly 3.67 times more likely to share misinformation, Fig 2 shows that misinformation–i.e., explicitly false claims about COVID-19 – was only present in a minority of posts. Thus, prior studies which used credibility as a proxy for misinformation may have *over*estimated the prevalence of explicitly false claims. Explicit falsehoods, although harmful, seem to be rare. To the extent that "more credible" sources shared misinformation, they did so to report on or, in some cases, attempt to debunk, it. Thus, contrary to the claim of an "infodemic" of *misinformation*, posts about COVID-19 included less misinformation than other health-related posts prior to the pandemic.

Our results demonstrate that the volume of low-credibility content is much lower than the volume of high-credibility content on Twitter and Facebook. However, small volumes of harmful content could still be problematic if they garner a disproportionately large number of engagements. We found that this was not the case. To the contrary, content from the highest-quality sources–government and academic websites–was shared more often, on average, on

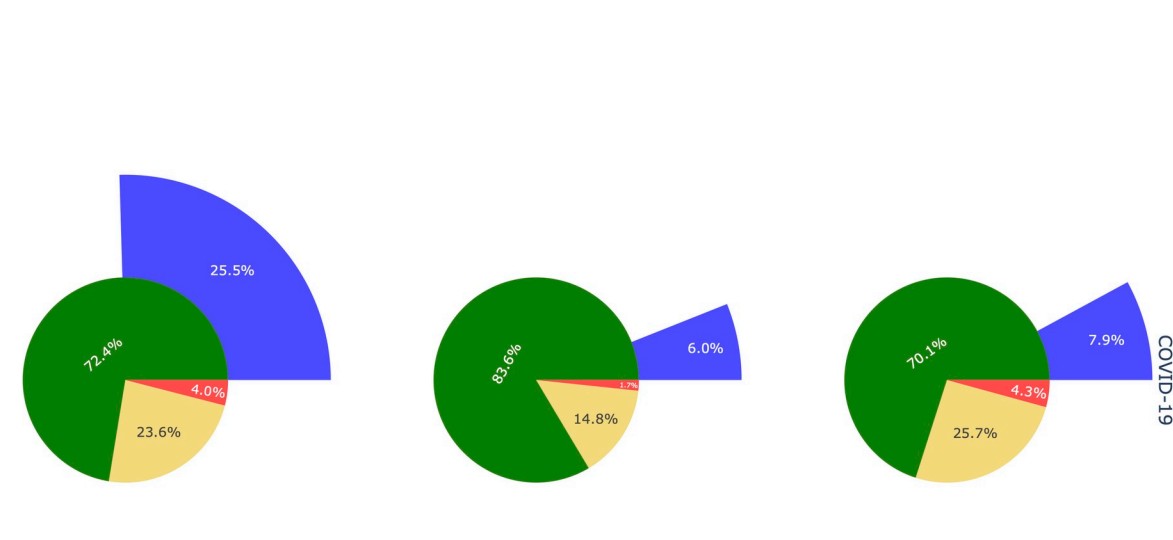

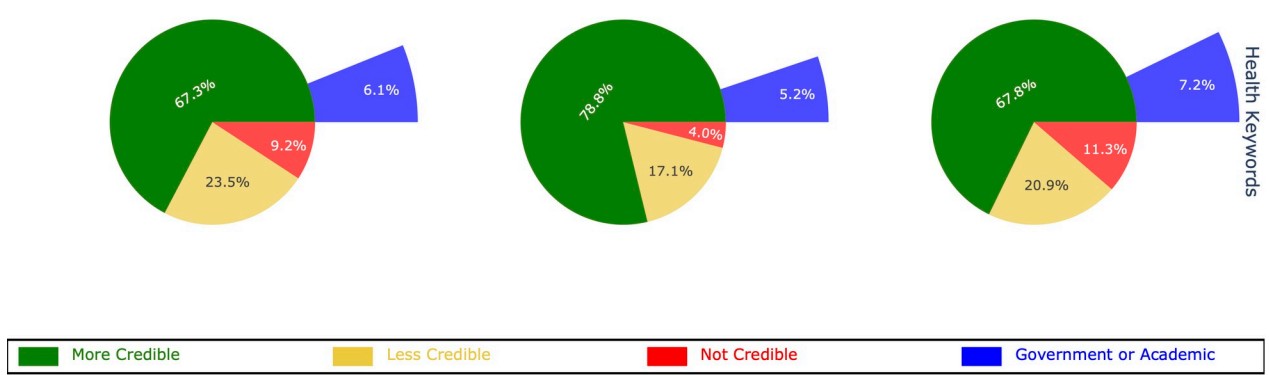

**Fig 2. Proportions of COVID-19 and health URLs for each credibility category and social media platform.**

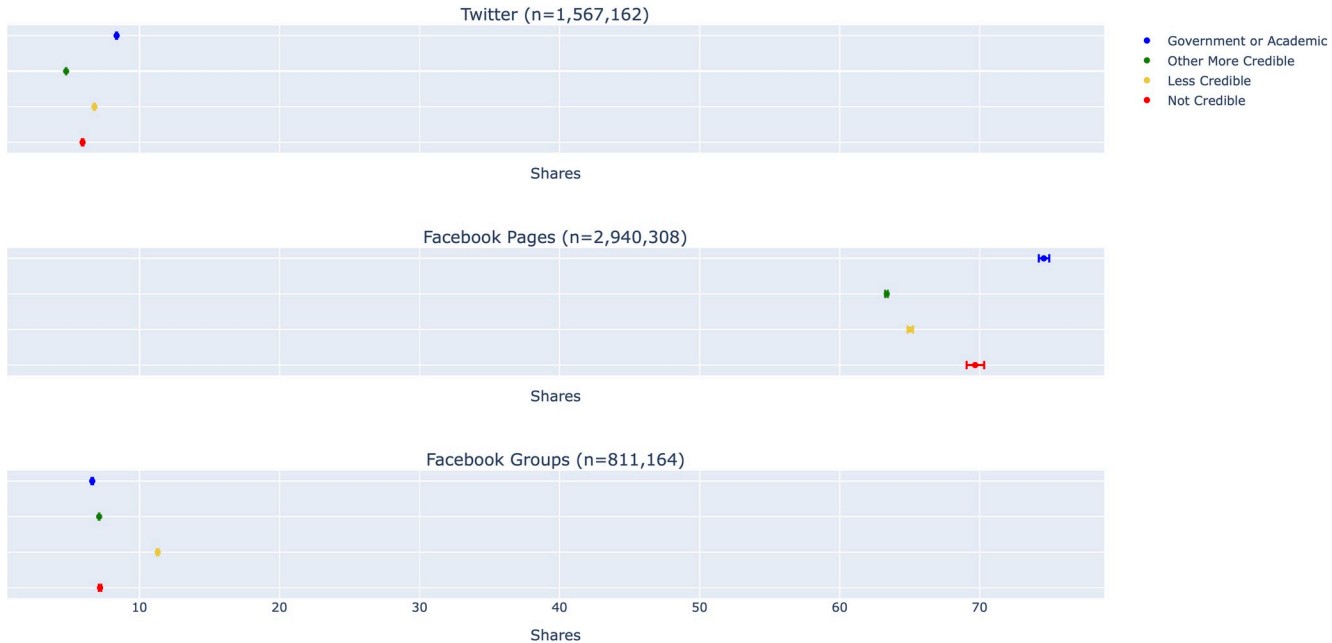

**Fig 3. Average number of shares for each credibility category by platform, estimated using negative binomial regression.**

both Twitter and Facebook. In Facebook Groups, where links to "not credible" sources were shared more often than links to high-quality sources, intermediate "less credible" sources were most frequently shared. However, we did not find that misinformation was significantly more prevalent in this category than in the "more credible" category.

Taken as a whole, these results suggest that misinformation about COVID-19 may largely be concentrated within specific online communities with limited spread overall. Online misinformation about COVID-19 remains problematic. However, our results suggest that the widespread reporting of false claims pertaining to COVID-19 may have been overstated at the start of the pandemic, whereas other health topics may be more prone to misinformation.

## Limitations

Our inclusion criteria for social media data are based on keywords associated with COVID-19, vaccine-preventable illnesses, and other health conditions. This collection procedure might introduce some noise in our dataset, for example if online actors exploited the virality of the COVID-19 hashtags/keywords to promote their content. If so, this would engender potentially more misinformation during the pandemic; in fact, we found that there was less (see S2 Fig, where we quantified proportions of "opportunistic" content). Furthermore, we used inclusion criteria that are comparable to prior studies, including those upon which the initial claim of an infodemic was based: a WHO/PAHO fact sheet from May 1, 2020 (https://iris.paho.org/bitstream/handle/10665.2/52052/Factsheet-infodemic_eng.pdf?sequence=14&isAllowed=y), defines the "infodemic" using keyword search terms that are similar to ours. Other studies of the "infodemic" have taken the same approach [12–14]. Thus, our findings contextualize previous work in this area which has primarily focused on low-credibility sources rather than a more holistic picture.

Our inclusion criteria yielded several unrated URLs, comprising roughly half our sample. These URLs were not primarily misinformative (see S3 Fig). However, even if unrated URLs did contain large quantities of misinformation, COVID-19 data were statistically significantly

less likely to contain this unrated content on all social media platforms studied compared to what would be expected prior to the pandemic.

## Conclusions

Taken together, our findings suggest that the "infodemic" is, in fact, a general feature of health information online, that is not restricted to COVID-19. In fact, COVID-19 content seems less likely to contain explicitly false facts. This does not mean that misinformation about COVID-19 is absent; however, it does suggest that attempts to combat it might be better informed by comparison to the broader health misinformation ecosystem. Such a comparison would potentially engender a more dramatic response.

Health leaders who have focused on COVID-19 misinformation should acknowledge that this problem affects other areas of health even more so. Beyond the COVID-19 infodemic, calls-to-action to address medical misinformation more broadly should be given higher priority.

## Supporting information

**S1 File Appendix A. Unshortening URLs.**
(PDF)

**S2 File. Appendix B.** Measuring source credibility.
(PDF)

**S3 File. Appendix C.** Categorizing tweets with multiple URLs.
(PDF)

**S1 Fig. Raw counts of posts and URLs in each dataset.** URLs are segmented by whether they were rated, unrated, or "in platform" (e.g., pointing from Facebook to Facebook or from Twitter to Twitter).
(PDF)

**S2 Fig. Content proportions in each dataset (n = 600 for each credibility category).**
(PDF)

**S3 Fig. Proportion of posts sharing information and also containing falsehoods ("misinformation") broken down by credibility category.** Error bars reflect one standard error.
(PDF)

## Author Contributions

**Conceptualization:** David A. Broniatowski, Daniel Kerchner.

**Data curation:** David A. Broniatowski, Daniel Kerchner, Fouzia Farooq, Xiaolei Huang, Mark Dredze.

**Formal analysis:** David A. Broniatowski, Daniel Kerchner, Fouzia Farooq, Amelia M. Jamison.

**Funding acquisition:** David A. Broniatowski, Sandra Crouse Quinn.

**Investigation:** David A. Broniatowski, Daniel Kerchner, Fouzia Farooq, Xiaolei Huang, Amelia M. Jamison, Mark Dredze.

**Methodology:** David A. Broniatowski, Daniel Kerchner, John W. Ayers.

**Project administration:** David A. Broniatowski, Mark Dredze, Sandra Crouse Quinn.

**Resources:** David A. Broniatowski.

**Software:** David A. Broniatowski, Daniel Kerchner, Fouzia Farooq, Xiaolei Huang.

**Supervision:** David A. Broniatowski, Mark Dredze, Sandra Crouse Quinn.

**Validation:** David A. Broniatowski, Daniel Kerchner, Fouzia Farooq, Amelia M. Jamison.

**Visualization:** David A. Broniatowski, John W. Ayers.

**Writing – original draft:** David A. Broniatowski, Fouzia Farooq.

**Writing – review & editing:** David A. Broniatowski, Daniel Kerchner, Fouzia Farooq, Xiaolei Huang, Amelia M. Jamison, Mark Dredze, Sandra Crouse Quinn, John W. Ayers.

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
