## [Decision Letter · Decision Letter 0]

2 Sep 2021

PONE-D-21-17260

Twitter and Facebook posts about COVID-19 are less likely to spread misinformation compared to other health topics

PLOS ONE

Dear Dr. Broniatowski,

Thank you for submitting your manuscript to PLOS ONE. After careful consideration, we feel that it has merit but does not fully meet PLOS ONE’s publication criteria as it currently stands. Therefore, we invite you to submit a revised version of the manuscript that addresses the points raised during the review process.

The paper needs a MAJOR REVISION in order to be evaluated for a publication. In particular, reviewers highlighted that the contribution of the paper needs to be improved. Please follow the suggestions included in this email.

We look forward to receiving your revised manuscript.

Kind regards,

Barbara Guidi

Academic Editor

PLOS ONE

Journal Requirements:

This work was supported in part by grant number R01GM114771 to D.A. Broniatowski and S.C. Quinn, and by the James S. and John L. Knight Foundation to the GW Institute for Data, Democracy, and Politics.

David A. Broniatowski received an honorarium from the United Nations Shot@Life Foundation – a non-governmental organization that promotes childhood vaccination. Mark Dredze holds equity in Sickweather Inc. and has received consulting fees from Bloomberg LP and Good Analytics Inc. None of these organizations had any role in the study design, data collection, and analysis, decision to publish, or preparation of the article. The remaining authors declare no competing interests.

Please respond by return email with your amended Competing Interests Statement and we will change the online submission form on your behalf.

Reviewers' comments:

Reviewer's Responses to Questions

**Comments to the Author**

1. Is the manuscript technically sound, and do the data support the conclusions?

Reviewer #1: Partly

Reviewer #2: Partly

2. Has the statistical analysis been performed appropriately and rigorously? 

Reviewer #1: Yes

Reviewer #2: Yes

3. Have the authors made all data underlying the findings in their manuscript fully available?

Reviewer #1: No

Reviewer #2: Yes

4. Is the manuscript presented in an intelligible fashion and written in standard English?

Reviewer #1: No

Reviewer #2: Yes

5. Review Comments to the Author

Reviewer #1: The paper provides a study of the credibility of the news propagated on Twitter and Facebook about Covid-19. The authors used the URLs to define if the news (posts, tweets) are credible or not. The results show that the news concerning a Covid-19 are more credible than other health topics.

It will more interesting if the authors could add another factor analyzing the text keywords or summary to measure the credibility of the text and not only the URL.

The citation should be put before the end of the sentence. “spurred an 5 “infodemic” of misinformation.(1)”

The paper should be formatted to respect the Plos One format.

Reviewer #2: In this paper, the authors present a study concerning misinformation spreading during the covid-19 pandemic and compare it with misinformation spreading regarding other health topics. The paper is well written and relatively easy to follow, but the contribution should be improved.

In detail, the authors have more than 318 million posts from Twitter and Facebook but only use 3000 for their analyses concerning misinformation spreading. I think the authors can improve on that. The 3000 manually annotated posts can be used as a basis, such as to train an AI classifier, and then use the classifier to classify the remaining posts in order to gain more insight concerning misinformation. on a much bigger dataset. For instance, do people share news from unreliable sources to "tag" these sources and make other people aware that fake news are circulating on certain topics? Or are people sharing links that support their thesis? The topic is interesting, but the contribution is not enough for a journal publication.

I also recommend the authors restyle Figure 1 because it's very hard to understand and some labels are impossible to read. Maybe use a pie chart or a stacked histogram? I think you should also discuss and describe the results shown in figures 2 and 3 in more detail.

6. PLOS authors have the option to publish the peer review history of their article (what does this mean?). If published, this will include your full peer review and any attached files.

Reviewer #1: No

Reviewer #2: No

---

## [Author Response · Author response to Decision Letter 0]

17 Oct 2021

COMMENT R1-1: The paper provides a study of the credibility of the news propagated on Twitter and Facebook about Covid-19. The authors used the URLs to define if the news (posts, tweets) are credible or not. The results show that the news concerning a Covid-19 are more credible than other health topics.

It will more interesting if the authors could add another factor analyzing the text keywords or summary to measure the credibility of the text and not only the URL.

RESPONSE R1-1: We thank the reviewer for this comment, because it allows us to clarify the main construct of our study: credibility. In our study, credibility is defined on URLs and is a feature of publishers; not of content. Several prominent, and highly-cited, papers have been published using this assumption:

• Lazer, D. M., Baum, M. A., Benkler, Y., Berinsky, A. J., Greenhill, K. M., Menczer, F., ... & Zittrain, J. L. (2018). The science of fake news. Science, 359(6380), 1094-1096.

• Grinberg, N., Joseph, K., Friedland, L., Swire-Thompson, B., & Lazer, D. (2019). Fake news on Twitter during the 2016 US presidential election. Science, 363(6425), 374-378.

• Pennycook, G., & Rand, D. G. (2019). Fighting misinformation on social media using crowdsourced judgments of news source quality. Proceedings of the National Academy of Sciences, 116(7), 2521-2526.

• Singh, L., Bode, L., Budak, C., Kawintiranon, K., Padden, C., & Vraga, E. (2020). Understanding high-and low-quality URL Sharing on COVID-19 Twitter streams. Journal of computational social science, 3(2), 343-366.

• Yang, K. C., Pierri, F., Hui, P. M., Axelrod, D., Torres-Lugo, C., Bryden, J., & Menczer, F. (2021). The COVID-19 Infodemic: Twitter versus Facebook. Big Data & Society, 8(1), 20539517211013861.

• Cinelli, M., Quattrociocchi, W., Galeazzi, A., Valensise, C. M., Brugnoli, E., Schmidt, A. L., ... & Scala, A. (2020). The COVID-19 social media infodemic. Scientific Reports, 10(1), 1-10.

Thus, the assumption that URLs are an adequate measure of credibility is widespread and accepted in top journals, including in Science, PNAS, and Scientific Reports. 

Using credibility as our primary metric, this paper makes a major contribution to the existing, published literature: we are the first one to compare COVID credibility scores to those of other health topics. Prior papers analyzing the credibility of the COVID infodemic studied COVID in isolation and so there was no comparator. Our novel contribution is that we are able to quantify whether the credibility of COVID information was higher or lower than information about other health topics. This adds much needed context to the debate.

Broadly speaking, we agree with the reviewer that credibility is only a proxy measure of misinformativeness. We therefore conducted an additional analysis where we rated the misinformation content of posts that differ in credibility (lines 173-184). This analysis demonstrates clearly that content with low credibility URLs is significantly more misinformative than other types of content. We quantified how much of this content is explicitly false. Overall, our results are robust – we found that COVID content has a lower proportion of low credibility URLs than other health topics. We now include discussion about these points in our manuscript (lines 263-271), and we thank the reviewer for encouraging this clarification. 

COMMENT R1-2: The citation should be put before the end of the sentence. “spurred an 5 “infodemic” of misinformation.(1)”

The paper should be formatted to respect the Plos One format.

RESPONSE R1-2: Per the reviewer’s recommendation we have put the referenced citation before the end of the sentence, and we have formatted the paper to respect the Plos One format. We thank the reviewer.

COMMENT R2-1: In this paper, the authors present a study concerning misinformation spreading during the covid-19 pandemic and compare it with misinformation spreading regarding other health topics. The paper is well written and relatively easy to follow, but the contribution should be improved.

RESPONSE R2-1: We thank the reviewer for the overall positive feedback and comments! 

COMMENT R2-2: In detail, the authors have more than 318 million posts from Twitter and Facebook but only use 3000 for their analyses concerning misinformation spreading. I think the authors can improve on that. The 3000 manually annotated posts can be used as a basis, such as to train an AI classifier, and then use the classifier to classify the remaining posts in order to gain more insight concerning misinformation. on a much bigger dataset. 

RESPONSE R2-2: Our primary aim was to assess misinformation by examining the credibility of URLs shared on Facebook and Twitter so that we might compare COVID-19 posts to posts about other health topics. As we state in RESPONSE R1-1, we drew upon a widely accepted method for doing so. Despite the widespread use of this technique, which has been published in top journals, we provided an additional validity check by annotating 3000 posts by hand. This validity check confirms that low-credibility URLs are more likely to contain misinformation than high-quality URLs. Thus, our inferences both rely upon a solid body of literature and provide additional data to support our claims.

Nevertheless, we agree with the reviewer that the 3000 manually annotated posts can be used as a basis to make more refined inferences about the larger dataset. Although the reviewer suggests training an AI classifier to make these inferences, we respectfully point out that our sample of 3000 posts was stratified for each credibility category with 600 posts drawn uniformly at random from within each category. According to the Law of Large Numbers, the mean proportions of each observed label in our dataset will converge to the corresponding proportions of these labels in our populations. Therefore, a machine learning classifier is unnecessary for us to make strong statistically valid inferences regarding the underlying structure of the dataset. Furthermore, any classifier that we train would introduce sources of systematic error and bias, with these errors and biases depending on the specific choice of classifier used. On the other hand, the statistical approach that we use – stratified randomly sampling – relies on distribution-free chi-square tests and is therefore guaranteed to be free of systematic errors (i.e., biases). Furthermore, this statistical approach allows us to explicitly quantify the variance error (i.e., random error, as opposed to systematic error) in our measures, as is reflected in the confidence intervals and relative risk ratios on lines 221-231. We thank the reviewer for encouraging us to clarify the rigor of our exposition. 

COMMENT R2-3: For instance, do people share news from unreliable sources to "tag" these sources and make other people aware that fake news are circulating on certain topics? Or are people sharing links that support their thesis? The topic is interesting, but the contribution is not enough for a journal publication.

RESPONSE R2-3: We thank the reviewer for encouraging us to more explicitly examine how content from different credibility categories are used. We have conducted a more in-depth analysis of our hand-annotated sample, which directly addresses the reviewer’s concern that people might be sharing news from unreliable sources to ‘tag’ these sources. We found that when people reported on questionable content, such as explaining conspiracy theories or reporting on claims that bleach cures COVID, they typically did so by referencing “more credible” and “less credible” sources. In fact, this was the principal source of misinformation discussed by “more credible sources”. In contrast, we did not find that sharing “not credible” content did so frequently. We now include this discussion in the manuscript (lines 259-262).

COMMENT R2-4: I also recommend the authors restyle Figure 1 because it's very hard to understand and some labels are impossible to read. Maybe use a pie chart or a stacked histogram? I think you should also discuss and describe the results shown in figures 2 and 3 in more detail.

RESPONSE R2-4: We have relabeled restyled Figure 1 as a pie chart. We also now discuss and describe the results shown in Figures 2 and 3 in more detail. We thank the reviewer for this suggestion.

In conclusion, we have addressed all of the reviewers’ comments to the best of our abilities and we believe that our paper has significantly improved as a result. We thank the reviewers for their attention to detail, and we look forward to a positive response.

---

## [Decision Letter · Decision Letter 1]

10 Dec 2021

Twitter and Facebook posts about COVID-19 are less likely to spread misinformation compared to other health topics

PONE-D-21-17260R1

Dear Dr. Broniatowski,

We’re pleased to inform you that your manuscript has been judged scientifically suitable for publication and will be formally accepted for publication once it meets all outstanding technical requirements.

Kind regards,

Barbara Guidi

Academic Editor

PLOS ONE

Additional Editor Comments (optional):

Reviewers' comments:

Reviewer's Responses to Questions

**Comments to the Author**

1. If the authors have adequately addressed your comments raised in a previous round of review and you feel that this manuscript is now acceptable for publication, you may indicate that here to bypass the “Comments to the Author” section, enter your conflict of interest statement in the “Confidential to Editor” section, and submit your "Accept" recommendation.

Reviewer #2: All comments have been addressed

2. Is the manuscript technically sound, and do the data support the conclusions?

Reviewer #2: Partly

3. Has the statistical analysis been performed appropriately and rigorously? 

Reviewer #2: (No Response)

4. Have the authors made all data underlying the findings in their manuscript fully available?

Reviewer #2: (No Response)

5. Is the manuscript presented in an intelligible fashion and written in standard English?

Reviewer #2: (No Response)

6. Review Comments to the Author

Reviewer #2: (No Response)

7. PLOS authors have the option to publish the peer review history of their article (what does this mean?). If published, this will include your full peer review and any attached files.

Reviewer #2: No

---

## [Editor Report · Acceptance letter]

28 Dec 2021

PONE-D-21-17260R1 

Twitter and Facebook posts about COVID-19 are less likely to spread misinformation compared to other health topics 

Dear Dr. Broniatowski:

I'm pleased to inform you that your manuscript has been deemed suitable for publication in PLOS ONE. Congratulations! Your manuscript is now with our production department. 

Kind regards, 

on behalf of

Dr. Barbara Guidi 

Academic Editor

PLOS ONE